# Prostate Cancer Detection in Bi-Parametric MRI using Zonal Anatomy-Guided U-Mamba with Multi-Task Learning

**Michael S. Larsen**[1,2]                                    MICHAEL.S.LARSEN@NTNU.NO
**Syed Farhan Abbas**[2]                                         SYED.F.ABBAS@NTNU.NO
**Tone F. Bathen**[2]                                          TONE.F.BATHEN@NTNU.NO
**Mattijs Elschot**[2,3]                                     MATTIJS.ELSCHOT@NTNU.NO
**Gabriel Kiss**[1]                                              GABRIEL.KISS@NTNU.NO
**Frank Lindseth**[1]                                                 FRANKL@NTNU.NO

[1] *Department of Computer Science, Norwegian University of Science and Technology*

[2] *Department of Circulation and Medical Imaging, Norwegian University of Science and Technology*

[3] *Central Staff, St. Olavs Hospital, Trondheim University Hospital*

**Editors:** Accepted for publication at MIDL 2025

## Abstract

Prostate cancer (PCa) remains a leading cause of cancer-related morbidity, emphasizing the need for accurate and non-invasive diagnostic tools. While deep learning models have advanced PCa detection in magnetic resonance imaging (MRI), they often fail to integrate anatomical knowledge. This study evaluates U-Mamba, a deep learning architecture designed to enhance long-range dependency modeling with linear time complexity, for PCa detection. Furthermore, a multi-task learning (MTL) extension, U-Mamba MTL, is introduced to incorporate prostate zonal anatomy, aligning with clinical diagnostic workflows. The models were assessed using diverse datasets, including the PI-CAI hidden tuning cohort (N=100) and an in-house collected out-of-distribution cohort (N=200). Results demonstrate that U-Mamba achieves state-of-the-art detection performance, while U-Mamba MTL further improves PCa detection through the auxiliary zonal segmentation task. These findings highlight the potential of integrating U-Mamba with anatomical context to improve PCa detection. The code and model weights are available at https://github.com/mokkalokka/U-MambaMTL.

**Keywords:** Deep Learning, Medical Image Analysis, Prostate Cancer, Mamba, Multi-Task Learning

## 1. Introduction

Prostate cancer (PCa) is the fourth most prevalent cancer worldwide, despite exclusively affecting the male population (Bray et al., 2024). Prostate biopsies remain the gold standard for classifying cancer aggressiveness; however, the procedure is associated with health risks (Borghesi et al., 2017). To minimize unnecessary biopsies, magnetic resonance imaging (MRI) is increasingly used alongside prostate-specific antigen (PSA) blood tests as an effective, non-invasive tool for detection of clinically significant PCa (csPCa) using the prostate imaging-reporting and data system (PI-RADS) v2.1 protocol (Park et al., 2021). However, the diagnostic accuracy of MRI assessment can vary significantly depending on the reader's level of expertise (Wei et al., 2021).

A promising solution to mitigate this inter-reader variability in prostate MRI assessment is the application of artificial intelligence (AI) for automatic PCa detection. Training and validating AI models, however, requires large amounts of labeled data. In response to this need, the organizers of the PI-CAI challenge provided a large-scale multi-center dataset comprising 10,207 bi-parametric MRI (bpMRI) cases to advance research in PCa detection (Saha et al., 2024). While only a small subset of this dataset (N=1,500) is publicly available for training and validation, it still surpasses the size of previously available labeled prostate bpMRI datasets (Adams et al., 2022; Litjens et al., 2017).

The top-performing submissions to the PI-CAI challenge utilized either convolutional neural network (CNN)-based architectures (Debs et al., 2022; Li et al., 2022; Karagoz et al., 2023) or hybrid CNN-transformer architectures (Yuan et al., 2022; Kan et al., 2022). The challenge organizers also introduced three strong baseline methods, leveraging three widely used CNN-based architectures. The first, nnU-Net, is a self-configuring network for medical segmentation that optimizes pre- and post-processing as well as architectural parameters based on the dataset and available computing resources (Isensee et al., 2021). The second, nnDetection, is similarly self-configuring but focuses on object detection using the Retina U-Net architecture (Baumgartner et al., 2021; Jaeger et al., 2020). The final baseline is a standard CNN-based U-Net (Ronneberger et al., 2015).

While CNN-based architectures dominate the field, they are inherently limited in capturing long-range dependencies due to the localized nature of convolutional filters. Transformer-based architectures, on the other hand, offer greater potential for modeling long-range dependencies but face challenges such as computational complexity, particularly in dense prediction tasks like segmentation, where small patch sizes and windowed self-attention are often required (Liu et al., 2021). These constraints reduce their ability to fully leverage long-range information.

A recent alternative to CNNs and transformers called Mamba (Gu and Dao, 2023), claims to excel at leveraging long range dependencies for sequence to sequence tasks while maintaining a linear time complexity. U-Mamba (Ma et al., 2024), is one of the most popular mamba adaptations for medical image segmentation tasks, which is reported to achieve state of the art segmentation performance. However, efficacy on PCa detection in bpMRI remains unknown.

The prostate comprises two main zones: the transitional zone (TZ) and the peripheral zone (PZ), with the PZ accounting for most PCa cases. In PI-RADS v2.1, the dominant MRI sequence is determined by the lesion's zone. While zonal segmentation in MRI using deep learning has been extensively studied (Adams et al., 2022; Kou et al., 2024; Cuocolo et al., 2021; Aldoj et al., 2020), most PCa detection methods overlook anatomical knowledge like prostate zones. Some PCa detection studies use zonal masks as inputs (Yuan et al., 2022; Karagoz et al., 2023), while (Zheng et al., 2024) included zones as output classes using mpMRI. While DCE (included in mpMRI) has been reported to improve PCa detection in certain populations, particularly in men of African descent (Zabihollahy et al., 2023), recent studies have demonstrated that bpMRI is non-inferior to mpMRI for general PCa diagnosis and is now commonly used as a more cost-effective and less time-consuming alternative (Twilt et al., 2024). The increasing adoption of bpMRI in clinical workflows supports its relevance for deep learning-based PCa detection.

We propose a zonal anatomy-guided multi-task learning (MTL) approach using U-Mamba (Ma et al., 2024), marking its first application to PCa detection in bpMRI. While MTL has been explored before, our work is the first to use zonal anatomy as auxiliary segmentation targets, leveraging U-Mamba's long-range dependency modeling and linear time complexity to improve lesion detection in bpMRI. We introduce two MTL strategies (Single-Decoder and Dual-Decoder) to incorporate anatomical priors, significantly improving lesion detection. While achieving zonal segmentation performance on par with inter-reader variability, our results show that integrating zonal masks enhances PCa detection, with U-Mamba MTL-Single ranking 23rd out of 424 on the PI-CAI leaderboard, underscoring its competitiveness. The strong performance on both tasks suggests that U-Mamba MTL could serve as a promising clinical decision support tool for PCa assessment.

## 2. Methodology

### 2.1. Datasets

The datasets utilized in this study include the training cohort (N=1500) (which incorporates the ProstateX dataset (Litjens et al., 2017)), and the hidden tuning cohort (N=100) of the PI-CAI dataset (Saha et al., 2022). 425 cases in the PI-CAI training cohort are confirmed histologically to have clinically significant PCa (csPCa), defined as grade group $\geq$ 2. Of the csPCa cases in the PI-CAI training cohort, 220 cases include human expert annotations, while the remaining csPCa cases are derived from the approach outlined in (Bosma et al., 2023). Transition zone (TZ) and peripheral zone (PZ) masks for the training subset were AI-generated using a standard nnUNet (Isensee et al., 2021), trained on the ProstateX subset of the PI-CAI training data (Yuan et al., 2022).

The study further incorporated an in-house dataset (N=200) from NTNU/St. Olavs hospital, Trondheim, Norway (Krüger-Stokke et al., 2021), along with the Prostate158 dataset (N=158) (Adams et al., 2022). Both datasets provide expert annotations for PCa and zonal anatomy. PCa annotations for the in-house cohort are defined in the same manner as those in the PI-CAI datasets, but Prostate158 includes grade group 1, and is thus excluded from the csPCa detection assessment in this study. A resident radiologist with at least two years of experience at St. Olav's Hospital, Trondheim, delineated the in-house dataset using ITK-SNAP software in collaboration with a senior radiologist with over ten years of experience in prostate MRI. Annotations encompassed all MRI-visible lesions classified by PI-RADS, histopathologically confirmed lesions from biopsy or radical prostatectomy, and zonal anatomy.

An overview of all datasets utilized in this study is provided in Table 1 and the clinical variables for each cohort can be seen in Appendix A. All datasets in this study include T2-weighted (T2W), apparent diffusion coefficient (ADC), and high b-value (HBV) diffusion-weighted images, collectively referred to as bpMRI.

### 2.2. Network Architecture

We implemented the U-Mamba architecture (Ma et al., 2024) to investigate the hypothesis that the enhanced long-range dependency capabilities of Mamba (Gu and Dao, 2023) will be beneficial for PCa detection. As the performance of the Enc and Bot variant is reported

| Dataset | Cases | Type | Annotations |
|---|---|---|---|
| In-House[1] | 200 | 3T mpMRI | PCa, Zonal |
| PI-CAI Training cohort | 1500 | 1.5T, 3T bpMRI | PCa[2], Zonal[3] |
| PI-CAI Hidden tuning cohort | 100 | 1.5T, 3T bpMRI | PCa |
| Prostate158 | 158 | 3T bpMRI | PCa, Zonal |

Table 1: Prostate cancer dataset information. [1] denotes that the dataset is contained within the PI-CAI hidden test set cohort, [2] denotes that a subsection of the labels are AI generated (N=200) and [3] denotes that all the masks are AI generated

to be similar, we opted for the Bot variant due to it's reduced computational complexity (Isensee et al., 2024).

The particular configuration of the U-Mamba architecture used in this paper consists of 7 convolution stages in the encoder and decoder, where each stage in encoder consists of 2 ($3 \times 3$) convolutions. The decoder consists of the upsampling blocks in addition to a residual block. The bottleneck consists of the mamba-based block called the U-Mamba block in addition to a residual block.

### 2.3. U-Mamba MTL

To investigate whether incorporating zonal masks (TZ and PZ) improves PCa prediction, we explored two multitask learning (MTL) strategies using U-Mamba as the base architecture. Since zonal and PCa masks are not mutually exclusive, they can be treated as separate tasks. When tasks are highly related, a shared-parameter strategy is typically more effective. Conversely, if they are less related, allocating more task-specific parameters may be beneficial. Determining the optimal balance, however, requires experimentation.

We define the two tasks as $T_0$ = PCa and $T_1$ = Peripheral Zone (PZ) and Transitional Zone (TZ) zonal masks. Our U-Mamba MTL architectures can then be formulated as:

$$
\begin{aligned}
\mathbf{z} &= f_{\text{enc}}(\mathbf{x}; \theta_{\text{enc}}), \\
\mathbf{y}_i &= f_{\text{dec}_i}(\mathbf{z}; \theta_{\text{dec}_i}), \quad \forall i \in \{1, \ldots, N\}
\end{aligned}
\tag{1}
$$

The first strategy, U-Mamba MTL-Dual, uses $N = 2$, meaning two decoder branches separately predict $y_{T_0}$ (PCa) and $y_{T_1}$ (zonal masks). The encoder is shared, learning a common representation, while each decoder branch captures task-specific features.

The second strategy, U-Mamba MTL-Single, uses $N = 1$, meaning a single decoder predicts both $y_{T_0}$ and $y_{T_1}$, sharing all parameters across tasks.

Aside from the additional decoder in U-Mamba MTL-Dual, both models maintain the same overall structure as the base U-Mamba network. A complete architectural overview is provided in Figure 1.

### 2.4. Loss Functions

The two different tasks we aim to predict with our U-Mamba MTL architectures observe very different characteristics, which can cause issues with convergence if not handled care-

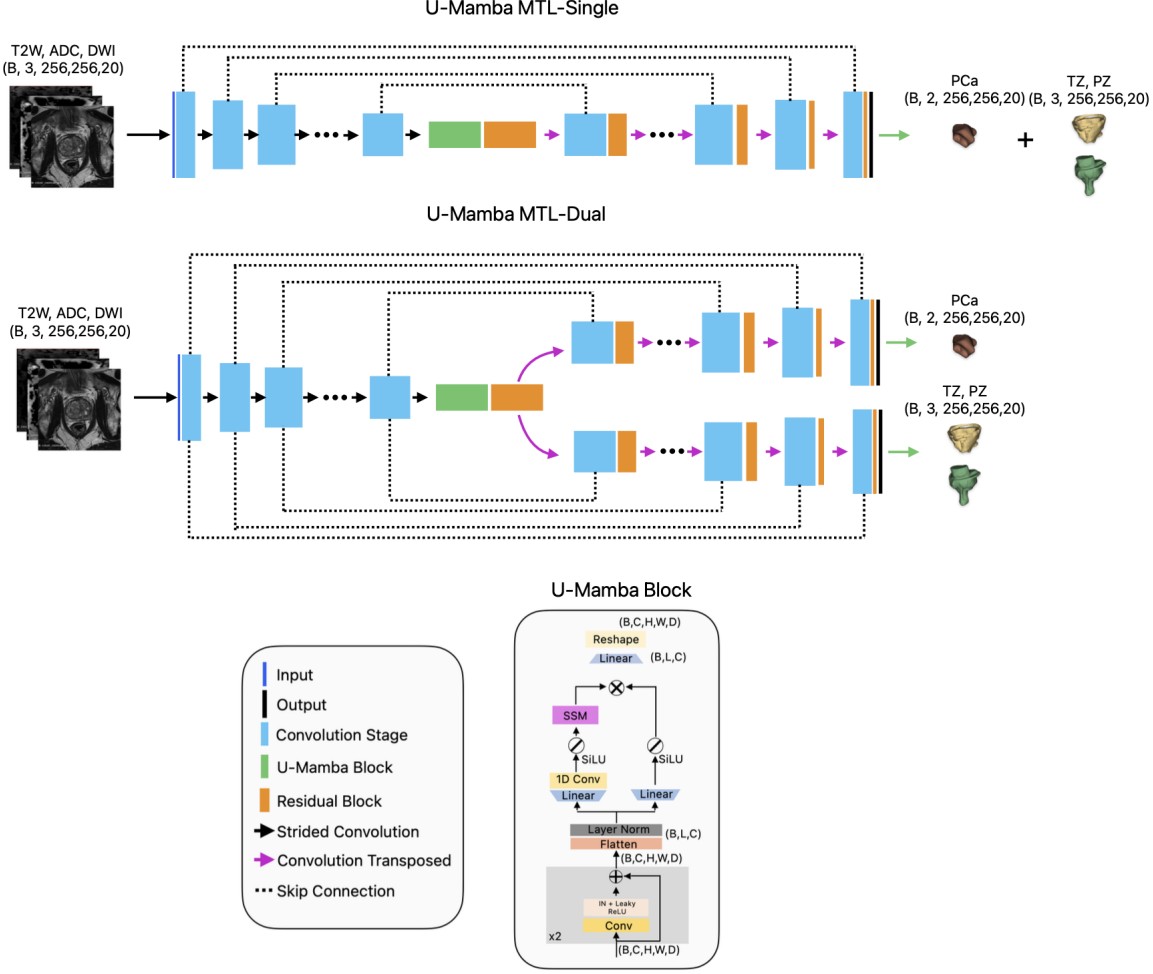

Figure 1: Architectural overview of U-Mamba MTL-Single and U-Mamba MTL-Dual

fully. The PCa task observes a severe class imbalance compared to the background, and is not present in all cases. These observations fits well with the selection criteria for the Focal loss function. The zonal mask prediction task on the other hand observes a moderate class imbalance compared to the background, and is present in all cases. Therefore, a combination of Dice and CE loss is deemed more suited for this task.

Due to the scale difference between the two task losses and the difference in relative difficulty of the tasks, a balancing factor $\beta$ is introduced. If we formulate the two targets in our MTL variants of U-Mamba as $T_0 = \text{PCa}$ and $T_1 = \text{PZ}$ and TZ zonal masks, the full formulation of the multi-task loss can be defined as:

$$\mathcal{L}_{T_0} = \mathcal{L}_{\text{Focal}}, \quad \mathcal{L}_{T_1} = \lambda\mathcal{L}_{\text{Dice}} + (1-\lambda)\mathcal{L}_{\text{CE}}, \quad \mathcal{L} = \mathcal{L}_{T_0} + \beta\mathcal{L}_{T_1}. \tag{2}$$

The weight balancing parameter $\lambda = 0.5$ which gives equal weight to the Dice and Cross Entropy component of $\mathcal{L}_{T_1}$. The weight balancing parameter $\beta$ is set to 0.2 to balance

both loss range and the relative difficulties of the tasks. Please note that the un-altered U-Mamba network uses $\mathcal{L}_{\text{Focal}}$ as its only loss function.

## 2.5. Model Training

Each model was trained using the PI-CAI challenge training dataset (N=1500) split into a training and a validation set by using 5-fold cross validation. Each split contains approximately 80% for training and 20% for validation.

All models were trained using 5-fold cross-validation for 200 epochs, a choice driven by observed early convergence during development, typically around 100 epochs. In contrast, the baseline models from the PI-CAI challenge organizers were trained for 1000 epochs. Training was conducted on a single A100 GPU (80GB VRAM) using a cosine annealing learning rate scheduler and the AdamW optimizer, producing five model weights per model. Final predictions for each model were generated using a mean ensemble across all model fold predictions.

To enhance the dataset diversity for model training, a set of data augmentations was used to augment the training data each epoch randomly. To ensure equal size of each image, we resample all images to the common spacing and perform crop or pad using the prostate as the center. Specific settings for each augmentation can be seen in Appendix B.

## 2.6. Baseline Models

To assess the performance of our model in relation to current state-of-the-art (SOTA) we opted to use the three baseline methods provided by the PI-CAI Challenge organizers which includes: nnUNet (Isensee et al., 2021), nnDetection (Baumgartner et al., 2021) and a standard U-Net (Ronneberger et al., 2015). In addition to the PI-CAI baselines we trained a SOTA transformer model called Swin UNETR (Hatamizadeh et al., 2022) using the same setup as the U-Mamba and our U-Mamba MTL model, except for the input size in the Z-dimension which was set to 32 due to model requirements.

## 2.7. Metrics

We assess PCa segmentations masks using average precision (AP) and area under the receiver operating curve (AUC), following PI-CAI guidelines (Saha et al., 2024). In order to compute the metrics, non-overlapping lesion candidates are extracted from the PCa probability map. The lesion candidates are iteratively extracted by selecting the voxel with the maximum probability and selecting all connected voxels with a minimum of 40% of its peak probability (Bosma et al., 2023). A PCa detection map is defined as the collection of all lesion candidates for a given case, where each lesion candidate have a single probability defined by its maximum probability.

A lesion is considered true positive in the AP calculation if its intersect over union exceeds 10%. AUC is computed per patient using the highest probability in the PCa detection map. The combined performance metric averages AP and AUC to evaluate lesion detection and patient-level PCa classification. The metrics are calculated with the picai_eval script provided by the PI-CAI challenge (Saha et al., 2024).

## 3. Results

In this section, two datasets from four separate institutions, the PI-CAI hidden tuning cohort (N=100) and our out-of-distribution in-house cohort (N=200), are used to evaluate the performance of our trained model against all baseline models. As each model was trained using 5-fold cross validation, a simple mean ensemble is employed on the softmax output of each model fold before extracting lesion candidates using the post processing steps described in Section 2.7. The lesion candidates are then reverted to original size before evaluating the metrics. Results for zonal segmentation can be seen in Appendix D.

### 3.1. Qualitative Results

Figure 2 shows a selection of qualitative results on the in-house dataset (N=200). The selected samples highlights cases where some or all models fails to produce correct predictions.

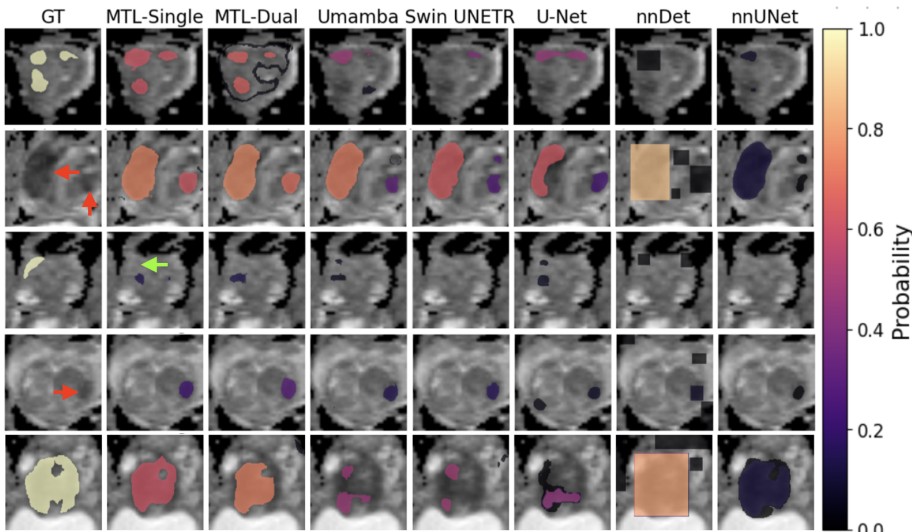

Figure 2: Qualitative comparison on the in-house dataset (N=200) of the PCa detection maps among all models compared to ground truth (GT), overlayed on ADC channel. Green arrow highlights an area where hypointensity is lacking. Red arrows highlights areas with hypointensity.

### 3.2. Quantitative results

The evaluation of the models on the PI-CAI hidden development set (N=100) was acquired by submitting a docker container for each trained model to the challenge website (Saha, 2025). Table 2 shows the quantitative results on the PI-CAI hidden development set, where our U-Mamba MTL-Dual achieved the highest aggregated score.

Table 3 shows the quantitative results on our in-house, out-of-distribution dataset (N=200). U-Mamba MTL achieves the highest score across all performance metrics.

| Model | Score | AUC | AP | Ranking | Parameters |
|---|---|---|---|---|---|
| **U-Mamba MTL Single (ours)** | **0.781** | 0.867 | **0.696** | **23rd** | 73.6M |
| U-Mamba MTL Dual (ours) | 0.735 | 0.843 | 0.622 | 134th | 114M |
| nnDetection | 0.734 | **0.885** | 0.582 | 139th | 24.7M |
| U-Net | 0.731 | 0.829 | 0.633 | 144th | 31.8M |
| U-Mamba | 0.727 | 0.820 | 0.635 | 157th | 73.6M |
| nnU-Net | 0.714 | 0.818 | 0.610 | 188th | 44.8M |
| Swin UNETR | 0.665 | 0.792 | 0.537 | 241st | 72.8M |

Table 2: Results on PI-CAI hidden development set (N=100)

| Model | Score | AUC | AP |
|---|---|---|---|
| **U-Mamba MTL Single (ours)** | **0.818±0.062** | **0.932 ± 0.041** | **0.705 ± 0.101** |
| U-Mamba MTL Dual (ours) | 0.805 ± 0.063 | 0.925 ± 0.043 | 0.685 ± 0.097 |
| U-Mamba | 0.799 ± 0.060 | 0.923 ± 0.044 | 0.674 ± 0.089 |
| Swin UNETR | 0.773 ± 0.063 | 0.902 ± 0.054 | 0.643 ± 0.094 |
| nnDetection | 0.765 ± 0.064 | 0.920 ± 0.045 | 0.610 ± 0.099 |
| U-Net | 0.759 ± 0.061 | 0.913 ± 0.043 | 0.605 ± 0.095 |
| nnUNet | 0.731 ± 0.070 | 0.910 ± 0.045 | 0.555 ± 0.113 |

Table 3: Results from our in-house dataset N=200, where ± refers to the largest difference from mean to the 95% confidence interval bounds, derived from 10.000 bootstrap samples (Jurdi et al., 2023).

## 4. Discussion and Conclusion

This work shows that our U-Mamba MTL-Single model outperformed baseline state-of-the-art models for PCa detection in bpMRI and achieved zonal segmentation performance comparable to inter-reader variability, as demonstrated on the Prostate158 dataset (Appendix D). Notably, it ranked 23rd out of 450 on the PI-CAI leaderboard, underscoring its competitiveness. Its strong AP and AUC scores indicate precise lesion localization and reliable patient-level classification, which are critical for guided prostate biopsy, improving targeting accuracy and reducing unnecessary procedures. Although the 95th confidence intervals are overlapping, both U-Mamba MTL variants achieved the highest scores on our out-of-distribution in-house dataset and the PI-CAI development dataset, demonstrating promising generalizability. These results highlight the importance of integrating zonal anatomy, which enhances PCa detection compared to using U-Mamba alone.

Analyzing the experimental results reveals that the single-decoder MTL approach significantly outperforms the dual-decoder MTL method. However, since PCa detection and zonal segmentation are fundamentally different tasks, an optimal balance of shared parameters may be achieved by partially splitting the decoder between the bottleneck and the prediction heads. Identifying this optimal point remains an avenue for future research.

All U-Mamba variants outperformed the baseline methods in terms of the combined score, except for the base U-Mamba on the PI-CAI hidden development dataset. This superior performance may be attributed to U-Mamba's enhanced ability to capture long-range dependencies, facilitated by the relatively large input size used in this study and the absence of patch-based learning. Additionally, its relatively high parameter count, comparable only to Swin UNETR, may have contributed to its effectiveness. However, further research is needed to confirm these factors' impact on performance.

Swin UNETR has architectural limitations, requiring a minimum input size of 32 for each dimension. Since the average Z-dimension size for the bpMRI data used in this study is 20, padding was necessary, which may have contributed to Swin UNETR's poor performance. While nnDetection demonstrated strong patient-level classification performance (AUC), its ability to localize PCa (AP) was among the lowest. This poor AP score is partly due to the nature of the nnDetection architecture, which only produces bounding boxes, unlike the other architectures in this study that generate segmentation masks. As AP is calculated by defining a lesion candidate as a true positive given a 10% overlap, the bounding-box-based approach may have been a limiting factor.

The qualitative results highlights challenges in PCa detection. Some false positives were caused by hypointense areas in the ADC channel, often indicative of PCa (rows 2 and 3, Figure 2). Conversely, a false negative occurred in a region lacking ADC hypointensity despite a ground truth annotation (row 3, Figure 2). Additionally, rows 1 and 3 illustrate improved PCa delineation in U-Mamba MTL-Single and Dual compared to the base U-Mamba, emphasizing the benefits of incorporating zonal anatomy context.

In conclusion, the U-Mamba architecture, with its enhanced long-range dependency modeling, improved PCa detection in bpMRI. Additionally, we demonstrated that integrating zonal masks via multi-task learning further enhanced PCa detection performance.

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

## Appendix A. Clinical Information

Tables 4, 5 and 6 contains the clinical information for the datasets used in this study and their origin.

| | PI-CAI Training cohort | | |
|---|---|---|---|
| | RUMC | ZGT | PCNN |
| Sites | 2 | 1 | 8 |
| Patients | 792 | 346 | 338 |
| Median age, years | 65 (60-69) | 67 (62-72) | 68 (63-72) |
| Median prostate-specific antigen, ng/mL | 9 (6-14) | 7 (5-11) | 9 (6-12) |
| Median prostate volume, mL | 63 (45-88) | 49 (36-70) | 50 (35-70) |
| Field strength, Tesla | 1.5, 3 | 3 | 1.5, 3 |
| Cases | 800 | 350 | 350 |
| Clinically significant prostate cancer (Gleason grade group $\geq 2$) | 236 (30%) | 80 (23%) | 109 (31%) |
| Positive MRI lesions | 614 | 186 | 287 |
| PI-RADS 3 | 149 (24%) | 32 (17%) | 65 (23%) |
| PI-RADS 4 | 226 (37%) | 71 (38%) | 141 (49%) |
| PI-RADS 5 | 239 (39%) | 83 (45%) | 81 (28%) |
| Gleason grade group 1 | 150 (36%) | 74 (45%) | 87 (43%) |
| Gleason grade group 2 | 136 (33%) | 46 (28%) | 78 (39%) |
| Gleason grade group 3 | 64 (16%) | 21 (13%) | 24 (12%) |
| Gleason grade group 4 | 28 (7%) | 6 (4%) | 7 (3%) |
| Gleason grade group 5 | 33 (8%) | 16 (10%) | 6 (3%) |

Table 4: Clinical variables and statistics for the PI-CAI Training cohort. Data presented as n, n (%), or median (IQR). Abbreviations: PCNN - Prostaat Centrum Noord-Nederland, PI-RADS - Prostate Imaging Reporting and Data System, PSA - prostate-specific antigen, RUMC - Radboud University Medical Center, ZGT - Ziekenhuisgroep Twente.

|  | PI-CAI Hidden tuning cohort | | |
|---|---|---|---|
|  | RUMC | ZGT | PCNN |
| Sites | 2 | 1 | 3 |
| Patients | 40 | 30 | 30 |
| Median age, years | 64 (58-70) | 66 (61-71) | 66 (60-74) |
| Median prostate-specific antigen, ng/mL | 8 (5-11) | 8 (6-11) | 9 (6-14) |
| Median prostate volume, mL | 64 (46-91) | 46 (35-54) | 42 (30-65) |
| Field strength, Tesla | 3 | 3 | 1.5, 3 |
| Cases | 40 | 30 | 30 |
| Clinically significant prostate cancer (Gleason grade group ≥ 2) | 16 (40%) | 12 (40%) | 13 (43%) |
| Positive MRI lesions | 21 | 25 | 33 |
| PI-RADS 3 | 4 (19%) | 3 (12%) | 7 (21%) |
| PI-RADS 4 | 10 (48%) | 7 (28%) | 17 (52%) |
| PI-RADS 5 | 7 (33%) | 15 (60%) | 9 (27%) |
| Gleason grade group 1 | 6 (24%) | 13 (52%) | 8 (35%) |
| Gleason grade group 2 | 8 (32%) | 7 (28%) | 8 (35%) |
| Gleason grade group 3 | 5 (20%) | 1 (4%) | 4 (17%) |
| Gleason grade group 4 | 2 (8%) | 1 (4%) | 2 (8%) |
| Gleason grade group 5 | 4 (16%) | 3 (12%) | 1 (4%) |

Table 5: Clinical variables and statistics for the PI-CAI Hidden tuning cohort. Data presented as n, n (%), or median (IQR). Abbreviations: PCNN - Prostaat Centrum Noord-Nederland, PI-RADS - Prostate Imaging Reporting and Data System, PSA - prostate-specific antigen, RUMC - Radboud University Medical Center, ZGT - Ziekenhuisgroep Twente.

|                                                    | In-House cohort |
|                                                    | STOH            |
|----------------------------------------------------|-----------------|
| Sites                                              | 1               |
| Patients                                           | 200             |
| Median age, years                                  | 66 (60-69)      |
| Median prostate-specific antigen, ng/mL            | 7 (5-12)        |
| Median prostate volume, mL                         | 50 (36-71)      |
| Field strength, Tesla                              | 3               |
| Cases                                              | 200             |
| Clinically significant prostate cancer (Gleason grade group $\geq$ 2) | 80 (40%)        |
| Positive MRI lesions                               | 131             |
| PI-RADS 3                                           | 29 (23%)        |
| PI-RADS 4                                           | 34 (25%)        |
| PI-RADS 5                                           | 68 (52%)        |
| Gleason grade group 1                              | 23 (18%)        |
| Gleason grade group 2                              | 40 (30%)        |
| Gleason grade group 3                              | 39 (30%)        |
| Gleason grade group 4                              | 14 (10%)        |
| Gleason grade group 5                              | 15 (12%)        |

Table 6: Clinical variables and statistics for the In-House cohort. Data presented as n, n (%), or median (IQR). Abbreviations: PI-RADS - Prostate Imaging Reporting and Data System, PSA - prostate-specific antigen, STOH - St. Olav's Hospital, Trondheim University Hospital.

## Appendix B. Data Augmentation

Table 7 contains the augmentations used for training and validation of the models used in this study. All augmentations were implemented using MONAI (Consortium, 2024).

| Augmentation | Parameter |
|---|---|
| Spacing* | $(0.5mm, 0.5mm, 3.0mm)$ |
| Crop or Pad* | $(256, 256, 20†)$ |
| Z-score normalization* | Channel wise |
| Random flip | Along each axis |
| Random Gaussian Smoothing | sigma=(0.5, 1.0) |
| Random Scale Intensity | 10% |
| Random Shift Intensity | 10% |
| Random Gaussian Noise | mean=0, std=0.1 |
| Random Affine | rotate=(0.15, 0.15, 0) |

Table 7: Dataset augmentations. * denotes that the augmentation is used for all splits, the rest is used only for the training split. † denotes that the Z dimension was changed to 32 for Swin UNETR as this is the lowest size for the architecture.

## Appendix C. Model Efficiency

Table 8 shows an efficiency analysis of each model used in this study.

| Model | Params | Training time | Inference time |
|---|---|---|---|
| U-Mamba MTL Single (ours) | 73.6M | 15H | 1.1s |
| U-Mamba MTL Dual (ours) | 114M | 16.5H | 0.9s |
| U-Mamba | 73.6M | 12.5H | 1.0s |
| Swin UNETR | 72.8M | 34H | 1.3s |
| U-Net | 31.8M | N/A | 1.3s |
| nnUNet | 44.8M | N/A | 31s |
| nnDet | 24.7M | N/A | 105s |

Table 8: Training time refers to each fold trained for 200 epochs on a A100 80GB VRAM GPU. The training time is not available for the PI-CAI baselines, as these models were trained by the PI-CAI organizers. Inference time includes the full inference pipeline (5 fold predictions, test time augmentations etc).

## Appendix D. Zonal Segmentation

Although the zonal segmentation task for our U-Mamba MTL models is deemed as an auxiliary task, these masks might be useful for downstream tasks given sufficient quality.

In order to assess the accuracy of the zonal masks, inference was performed on the 200 patients from the in-house dataset and the 158 patients from the Prostate158 dataset. The predicted segmentation masks are then compared to the ground truth using the Dice Score (DSC) metric (Figure 3). Please note that the DSC is compared to reader 1 in the P158 dataset.

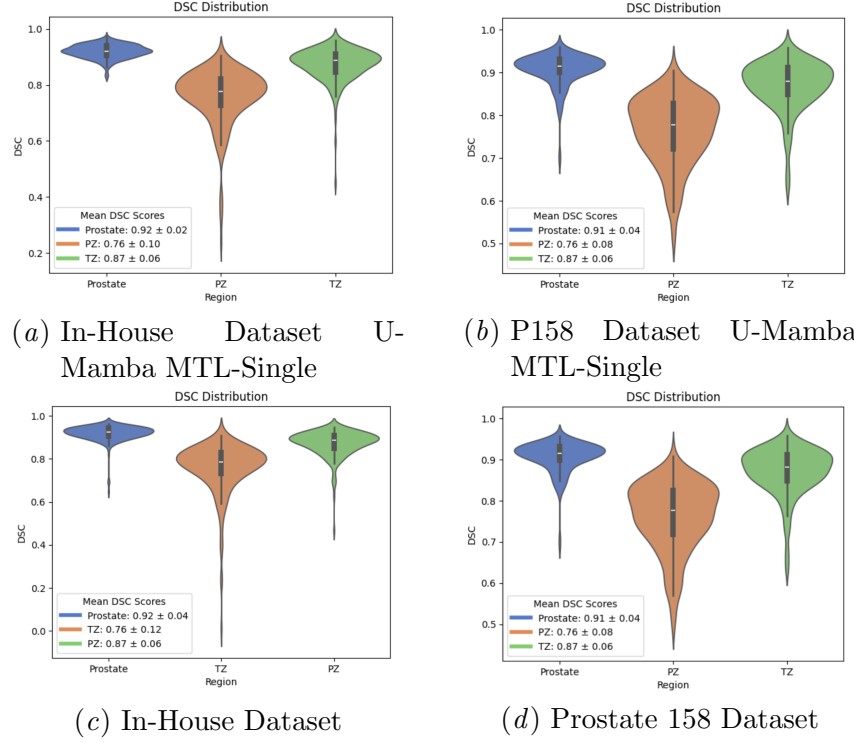

(a) In-House Dataset U-Mamba MTL-Single

(b) P158 Dataset U-Mamba MTL-Single

(c) In-House Dataset

(d) Prostate 158 Dataset

Figure 3: Dice score distributions for prostate zones for the U-Mamba MTL model

The auxiliary task of zonal segmentation within the both U-Mamba MTL architectures yielded strong results on both our in-house dataset and the Prostate158 dataset. Specifically, both our models achieved DSC scores of 0.76 and 0.87 for the peripheral zone (PZ) and transition zone (TZ), respectively, aligning closely with reported inter-reader variability ($DSC_{PZ} = 0.75$, $DSC_{TZ} = 0.87$). Notably, except for the ProstateX subset (N=346) of the PI-CAI Training set ( N=1500), all zonal masks were AI-generated, indicating that the model's zonal segmentation performance is largely a product of weak supervision.

