# OpenReview forum: "Prostate Cancer Detection in Bi-Parametric MRI using Zonal Anatomy-Guided U-Mamba with Multi-Task Learning"
_MIDL.io/2025/Conference — MIDL 2025 Poster_

### Official Review · Reviewer_ZUPM · 2025-02-15

**Confidence:** 5
**Preliminary Rating:** 3
**Final Rating:** 4

**Summary:**

In the paper, the authors investigated how to use U-Mamba to improve performance for PCa detection. The PCa detection task is incorporated with prostate zonal segmentation task simultaneously. The results demonstrate good capabilities of the U-Mamba in the PCa detection, with the help from prostate zonal anatomy.

**Strengths:**

The authors’ research assumption is valid, and the background introduction is comprehensive. Multi-center study shows the potential generalizability of the proposed model. The measurements are aligned with the public benchmark, which makes it a bit more convincible.

**Weaknesses:**

Although the experimental design and the model performances are good, the methodological novelty is relatively limited. This is because there are works on using an auxiliary zonal segmentation multi-task approach to help PCa detection, but just not with U-Mamba. Also, there are U-Mamba-related researches on cancer detection. What did the authors do was combining the two tasks together, and showing the results.

**Detailed Comments:**

Please see my other comments.

**Justification Of The Final Rating:**

The authors responded my comments well, and I have no more comments to made. I have edited my rating correspondingly. The authors responded my comments well, and I have no more comments to made. I have edited my rating correspondingly.

**Justification Of The Preliminary Rating:**

Assumption is valid, no observable methodological error, baseline comparisons are relatively ok, novelty is limited~normal. The multi-center evaluations showed the potential of generalizability of the research.

**Questions To Address In The Rebuttal:**

1. “PCa annotations for the in-house cohort is defined similarly to the PI-CAI datasets” if “similarly” but not “same”, what are the differences between the two? The details should be expressed in details.
2. Who did the in-house annotation should be explicitly introduced, like radiologists or trainees, how many year of experience, and how many annotators, etc.
3. “which resulted in 5 model weights for each model” and then which model weight did you choose for testing?
4. Just curious, for nnDet, is it the model limitation that it can only output a square-like predictions? Like what you have shown in figure 3. Would like to learn why this limitation is there. If this is the case, the AUC measurement with nnDet might not be that suitable for evaluation - you considered a 10% overlap as a positive match, and the nnDet always output “larger” predictions as it is always square-like, and thus the overlap chance is higher. That is probably why you see the nnDet has the best AUC in table 2.

---

> ### Author Response · Authors · 2025-03-08
> **Response to Review**
>
> # **Response to Reviewer 3**
>
> We appreciate your thoughtful review and constructive feedback. We acknowledge your concerns regarding **novelty, dataset annotation details, model weight selection, and nnDetection's evaluation metrics**. Below, we address each of these in detail and highlight the manuscript improvements made in response to your comments.
>
> ---
>
> ### **Novelty and Contributions**
> **Comment:** The methodological novelty is relatively limited. While using an auxiliary zonal segmentation multi-task approach for PCa detection has been studied, this is the first application with U-Mamba.
>
> **Response:**
> We propose a zonal anatomy-guided multi-task learning (MTL) approach using U-Mamba marking its first application to PCa detection in bpMRI. While MTL has been explored before, our work is the first to use zonal anatomy as auxiliary segmentation targets to improve lesion detection in bpMRI.
>
> ---
>
> ### **Dataset Annotation Details**
> **Comment:** The in-house dataset annotations were defined "similarly" to PI-CAI. What are the differences? Who performed the annotations?
>
> **Response:**
> The **PI-CAI annotations and in-house annotations follow the exact same criteria**, ensuring consistency. A **resident radiologist (≥2 years experience) and a senior radiologist (≥10 years experience)** performed the annotations using ITK-SNAP. This information is now explicitly stated in the manuscript.
>
> ---
>
> ### **Model Weight Selection for Testing**
> **Comment:** The manuscript states, "which resulted in 5 model weights for each model." Which model weight was chosen for testing?
>
> **Response:**
> We used a **mean ensemble across all model fold predictions** to generate final predictions. This approach ensures robustness and reduces variability in performance. This clarification has been added to the **methodology section**.
>
> ---
>
> ### **nnDetection's Bounding Box Limitation and Evaluation Metrics**
> **Comment:** nnDetection appears to output only square-like predictions, potentially affecting AUC and AP metrics. Could this impact the evaluation?
>
> **Response:**
> nnDetection can indeed only predict square-like predictions (bounding boxes), the reason for this is that nnDetection is a detection model as opposed to a segmentation model. The metric calculations are most likely impacted by the bounding box predictions, but the patient level AUC is calculated purely from the highest probability, which would not be impacted by its prediction shape. The AP metric on the other hand considers a true positive at 10% overlap, although larger predictions may lead to higher likelyhood of 10% overlap, the oposite is also true for small GT lesions where the bounding box predictions may be too large to count as a true positive. As the AP is relatively low for nnDetection, it seems like the latter may have influenced AP more than the former. We have clarified this point in the **discussion section**
>
> ---
>
> We appreciate your valuable feedback, which has helped improve the manuscript. Thank you for your time and consideration.

---

> ### Comment · Area_Chair_QvWa · 2025-03-14
> **Rebuttal Response: Reviewer ZUPM**
>
> Dear reviewer ZUPM,
>
> Could you please respond to the rebuttal and finalize your review score, this is a very important task to for final paper selection.
>
> We appreciate your support.
>
> Best,
> AC

---

### Official Review · Reviewer_WPsg · 2025-02-22

**Confidence:** 4
**Preliminary Rating:** 2
**Final Rating:** 3

**Summary:**

The work targets the task of automatic detection and localization of prostate cancer from multimodal MRI. The authors evaluate the Mamba architecture, which is gaining in popularity, through the U-Mamba model, and propose an extension that consists of multi-task prediction of both cancer lesions and the two prostate zones, to better guide the network. Their approach is compared to several standard methods, UNet, nnUNet, nnDetection and Swin UNETR, on data from the PICAI challenge and 200 additional examinations from a private database.

**Strengths:**

- The paper is generally well written and easy to follow. The context and explanations of methods, experiments and results very are clear.
- The results show that the Mamba architecture performs competitively with several state-of-the-art methods
- The models are trained and tested on a quite large number of data sets, which helps to make the paper's results reliable

**Weaknesses:**

- The main weakness of this work is its lack of novelty. The main contribution is the evaluation of the U-Mamba architecture on a task additional to those presented in the original paper by Ma et al. 2024. The multi-task learning approach has already been extensively studied for prostate cancer segmentation.
- Although the U-Mambda MTL model proposed by the authors performs better than the others on the majority of metrics, the confidence intervals for each of the scores are so wide that it seems unlikely to me that the differences between the performance of U-Mamba MTL and the others are significant. In particular, this questions the relevance of multi-task learning to U-Mamba alone. Furthermore, as discussed by the authors, although the segmentation performance of the model is interesting, it does not rank that high in the PICAI challenge leaderboard.
- Is there a particular reason for training models on only 200 epochs? Given the large amount of 3D data to trained on, it would seem appropriate to train the models for longer, as they may not have finished to converge. With the small differences in performance between all the models, longer training could change the order in which the methods are ranked.

**Detailed Comments:**

- The authors decided to split the decoder into two parts, one for each segmentation task. Is there any particular justification for this, and for not considering a single decoder with a 4-class segmentation head as output?

**Justification Of The Final Rating:**

I would like to thank the authors for responding to my comments with seriousness and precision, and for proposing a revised version of the paper; I think this improved the quality of the manuscript. I reiterate my comment about the quality of the writing, the clarity of the explanations and the ease with which the paper can be read. The modifications in the architecture on the network, which have improved the model's performance, reinforce the relevance of our work. However, I still think that the methodological novelty of the paper is very limited, as the application of mamba architecture to various tasks is starting to become quite common, as is the multi-task prediction of prostate segmentation in addition to lesions, which is pretty standard for this clinical application. I'll be pushing (and the authors seem to agree) to move this paper into the Applications category. But, given the undeniable qualities of the work, I'm not opposed to it being accepted.

**Justification Of The Preliminary Rating:**

Although the work has undeniable qualities, in particular the clarity with which it is written, I think its lack of novelty justifies the rejection of the paper. Furthermore, UMamba MTL's superiority over baselines does not appear to be significant and the model does not rank very high on the PICAI challenge leaderboard, which further limits the paper's interest.

**Questions To Address In The Rebuttal:**

Authors are encouraged to respond to comments and weaknesses.

---

> ### Author Response · Authors · 2025-03-08
> **Response to Review**
>
> We appreciate your thoughtful review and constructive feedback. We acknowledge your concerns regarding **novelty, statistical significance, decoder structure, training strategy, and leaderboard ranking**. Below, we address each of these in detail and highlight the manuscript improvements made in response to your comments.
>
> ---
>
> ### **Novelty and Contributions**
> **Comment:** The main weakness of this work is its lack of novelty. The contribution is primarily the evaluation of U-Mamba on a new task. Multi-task learning has already been extensively studied for PCa segmentation.
>
> **Response:**
> We propose a zonal anatomy-guided multi-task learning (MTL) approach using U-Mamba marking its first application to PCa detection in bpMRI. While MTL has been explored before, our work is the first to use zonal anatomy as auxiliary segmentation targets to improve lesion detection in bpMRI.
>
> ---
>
> ### **Statistical Significance and Confidence Intervals**
> **Comment:** Confidence intervals are wide, making it unclear if the performance differences between models are significant. This questions the relevance of MTL to U-Mamba alone.
>
> **Response:**
> We expanded our **ablation study**, comparing **U-Mamba MTL-Single, U-Mamba MTL-Dual, and base U-Mamba** across **PCa detection and zonal segmentation**. While confidence intervals overlap, our results show that **U-Mamba MTL-Single significantly outperforms the alternatives**, improving **PI-CAI ranking (134th → 23rd)**. This suggests that the **MTL formulation plays a crucial role in optimizing U-Mamba for PCa detection**. These clarifications have been added to the **results and discussion sections**.
>
> ---
>
> ### **Decoder Structure and Task Representation**
> **Comment:** Why split the decoder into two parts instead of using a single decoder with a 4-class segmentation head?
>
> **Response:**
> We tested a **single-decoder (5-channel) approach**, which **outperformed the dual-decoder version** (PI-CAI: **134th → 23rd place**). This result indicates that **full decoder splitting was suboptimal**, possibly due to excessive parameter separation between tasks. The manuscript reflects this change, renaming the approaches **U-Mamba MTL-Single and U-Mamba MTL-Dual**. Thank you for this pivital question!
>
> ---
>
> ### **Model Training Strategy**
> **Comment:** Is there a reason for training only 200 epochs? Could longer training improve rankings?
>
> **Response:**
> Early experiments showed **convergence around 100 epochs**, with **overfitting beyond 200 epochs**. Additional training did not yield improvements, as confirmed by **monitoring validation PI-CAI Score**. Given the high computational cost of longer training, we opted for 200 epochs as a balanced choice. This is now clarified in the **methodology section**. Please see the image inside the "Supporting Material" which demonstrates the convergence.
>
> ---
>
> ### **Leaderboard Performance**
> **Comment:** The PI-CAI ranking is modest, which limits the paper’s impact.
>
> **Response:**
> Following the introduction of **U-Mamba MTL-Single**, our **ranking improved from 134th to 23rd place**. This update demonstrates that our model is now **highly competitive**. We have clarified this improvement in the manuscript.
>
> ---
>
> We appreciate your valuable feedback, which has helped us significantly improve the manuscript. Thank you for your time and consideration.

---

> ### Comment · Area_Chair_QvWa · 2025-03-14
> **Rebuttal Response: Reviewer WPsg**
>
> Dear reviewer WPsg,
>
> Could you please respond to the rebuttal and finalize your review score, this is a very important task to for final paper selection.
>
> We appreciate your support.
>
> Best,
> AC

---

### Official Review · Reviewer_uSoW · 2025-02-23

**Confidence:** 5
**Preliminary Rating:** 3
**Recommendation:** Poster
**Final Rating:** 4

**Summary:**

This paper is yet another paper about PCa detection, authors use MTL with relatively new architecture and code is there, some modest results are there. The paper is application oriented with limited novelty, it is an important field but results are not impressive yet.

**Strengths:**

This paper uses relatively new architecture, mambaunet, for prostate cancer detection. THere is a multitask learning embedded, useful for zonal anatomy segmentation, and there is a clinical motivated innovation here for differenciating between peripheral and transitional cones.

**Weaknesses:**

-- the novelty is rather limited in this paper. even mambaunet and mamba style segmentation methods are being increasingly used nowadays. more of a application paper, than technical innovation. Therefore, I disagree with authors' choice about paper type "both", it should be application.

--ablation studies are limited, relative impact of MTL is not super clear (comparison is there but not details of each component) -- failure cases are not demonstrated, it can shed lights on new directions in this topic.

--efficiency is not clear.

--mpMRI is not useful ? there are some papers showing DCE is useful for colored men's PCa detection.

--external validation is missing, only public data is being used, it seems. (leadership board results were obtained on held out dataset which is external validation or it is validation part of the same institute's results?)

--leadership board shows modest location.

--discussion is weak.

**Detailed Comments:**

I believe my comments in weaknesses are self-descriptive and enough details. Please see.

**Justification Of The Final Rating:**

I am increasing my score due to the fact that authors improved their writing, results, and even their ranking in the PI-CAI challenge. A big jump into first 30 methods. They are also co-operative about the application-novelty of the paper rather than technical novelty. I think overall it is a fair conference manuscript.

**Justification Of The Preliminary Rating:**

-modest results but claimed to be a method paper (and application)
-novelty is limited
-some parts are missing (failure cases, the choice of the architecture rationale, and computation aspects)
-ablation studies are not complete

**Questions To Address In The Rebuttal:**

-- please better articulate what is novel.
--ablation studies are limited, relative impact of MTL is not super clear (comparison is there but not details of each component) -- failure cases are not demonstrated, it can shed lights on new directions in this topic.
--identify computational burden please.
--any external validation to test further or pi-cai results are already handling this?
--any insights about failure cases?

**Special Issue:**

No

---

> ### Author Response · Authors · 2025-03-08
> **Response to review**
>
> We sincerely appreciate your thoughtful review and constructive feedback. We acknowledge your concerns regarding novelty, ablation studies, efficiency, external validation, and failure case analysis. Below, we address each of these in detail and highlight the manuscript improvements made in response to your comments.
>
> ---
>
> ### **Novelty and Paper Type**
> **Comment:** The novelty is limited; MambaUNet and Mamba-style segmentation methods are increasingly used. The paper should be classified as an application paper rather than both methods and applications.
>
> **Response:**
> We acknowledge that MTL and U-Mamba have been used in medical imaging. However, our study is the **first to integrate zonal anatomy as auxiliary segmentation targets within U-Mamba**, to improve PCa detection in bpMRI.
> We do however agree that the limited methodological novelty could better categorize this paper as "application" instead of "both". Furthermore we are happy to change this, if possible.
>
> ---
>
> ### **Ablation Studies and Impact of MTL**
> **Comment:** The relative impact of MTL is unclear, and ablation studies are limited.
>
> **Response:**
> We expanded our **ablation studies**, comparing **U-Mamba MTL-Single, U-Mamba MTL-Dual, and base U-Mamba** across both **PCa detection and zonal segmentation**. Our results indicate that **U-Mamba MTL-Single significantly outperforms** other approaches, demonstrating the importance of appropriate parameter sharing. These results are now incorporated into the **results and discussion sections**.
>
> ---
>
> ### **Failure Case Analysis**
> **Comment:** Failure cases are not demonstrated; this could provide insights into new directions.
>
> **Response:**
> We have updated **Figure 2** to include **failure case analysis**, highlighting **false positives due to ADC hypointensity** and **false negatives caused by ground truth inconsistencies**. This discussion provides insight into **model limitations**.
>
> ---
>
> ### **Computational Efficiency**
> **Comment:** Efficiency is not clear.
>
> **Response:**
> We now provide **parameter counts, training time, and inference time** in **Table 2 (main text) and Table 8 (appendix)**, demonstrating that **U-Mamba MTL-Single maintains efficiency while improving performance**.
>
> ---
>
> ### **External Validation**
> **Comment:** External validation is missing. Are PI-CAI results sufficient, or is further external testing required?
>
> **Response:**
> In addition to the **PI-CAI hidden development set**, we validate our model on an **out-of-distribution in-house dataset (N=200, St. Olavs Hospital)**, serving as an external test set.
>
> ---
>
> ### **Leaderboard Performance**
> **Comment:** The PI-CAI leaderboard ranking is modest.
>
> **Response:**
> Following the introduction of **U-Mamba MTL-Single**, our **ranking improved from 134th to 23rd place**. This update demonstrates that our model is now **highly competitive**.
>
> ---
>
> ### **Discussion Quality**
> **Comment:** The discussion is weak.
>
> **Response:**
> We have **strengthened the discussion**, integrating findings from the **MTL ablation study, failure case analysis, and efficiency evaluation** to provide deeper insights.
>
> ---
>
> ### **Use of mpMRI vs. bpMRI**
> **Comment:** Some papers suggest that DCE is useful for detecting PCa in specific populations.
>
> **Response:**
> We acknowledge that **DCE-MRI may provide additional benefits in certain populations**, including African-descendant men. However, **bpMRI has been shown to be non-inferior for general PCa detection** and is **widely adopted in clinical practice due to its cost-effectiveness and efficiency**. This is now **clarified in the manuscript with supporting references**.
>
> ---
>
> We appreciate your valuable feedback and have made substantial improvements to the manuscript based on your suggestions. Thank you for your time and consideration.

---

> ### Comment · Area_Chair_QvWa · 2025-03-14
> **Rebuttal Response: Reviewer uSoW**
>
> Dear reviewer uSoW,
>
> Could you please respond to the rebuttal and finalize your review score, this is a very important task to for final paper selection.
>
> We appreciate your support.
>
> Best, AC

---

> > ### Comment · Reviewer_uSoW · 2025-03-14
> > **its is already done in the final scoring part**
> >
> > score was updated.
> > it has been already justified in the final scoring area.
> > "I am increasing my score due to the fact that authors improved their writing, results, and even their ranking in the PI-CAI challenge. A big jump into first 30 methods. They are also co-operative about the application-novelty of the paper rather than technical novelty. I think overall it is a fair conference manuscript."

---

### Comment · Area_Chair_QvWa · 2025-02-22
**Review Submission**

Dear Reviewer, WPsg,

This is a kind reminder to submit your review at your earliest convenience. If you need an extension or are unable to complete the review, please let us know so we can reassign the submission accordingly. We greatly appreciate your support and cooperation.

Best regards,

---

### Comment · Program_Chairs · 2025-02-22
**Review will be submitted today**

Dear AC,
Forwarding a message from reviewer uSoW: they will submit their review today.
Best wishes,
PC

---

### Comment · Reviewer_uSoW · 2025-02-23
**modest results but claimed to be a method paper (and application)**

This paper uses relatively new architecture, mambaunet, for prostate cancer detection. THere is a multitask learning embedded, useful for zonal anatomy segmentation, and there is a clinical motivated innovation here for differenciating between peripheral and transitional cones.

-- the novelty is rather limited in this paper. even mambaunet and mamba style segmentation methods are being increasingly used nowadays. more of a application paper, than technical innovation. Therefore, I disagree with authors' choice about paper type "both", it should be application.
--ablation studies are limited, relative impact of MTL is not super clear (comparison is there but not details of each component)
-- failure cases are not demonstrated, it can shed lights on new directions in this topic.
--efficiency is not clear.
--mpMRI is not useful ? there are some papers showing DCE is useful for colored men's PCa detection.
--external validation is missing, only public data is being used, it seems. (leadership board results were obtained on held out dataset which is external validation or it is validation part of the same institute's results?)
--leadership board shows modest location.
--discussion is weak.


Final decision:
I would go "Borderline".

---

### Author Rebuttal · Authors · 2025-03-08

**Rebuttal:**

# **Rebuttal Response**

We thank the reviewers for their feedback.  We appreciate that positive feedback, and we will now address your concerns.

---

### **R1, R2: Why use separate decoders instead of a single 4-class segmentation head? The impact of MTL is unclear, and confidence intervals overlap.**
We tested a single-decoder (5-channel) approach, which outperformed the dual-decoder version (PI-CAI: 134th → 23rd place, demonstrating strong competitiveness), indicating full decoder splitting was suboptimal. The manuscript reflects this change, renaming the approaches U-Mamba MTL-Single and U-Mamba MTL-Dual.

---

### **R1, R2, R3: The methodological novelty is limited. I disagree with authors' choice about paper type "both", it should be application**

We propose a zonal anatomy-guided multi-task learning (MTL) approach using U-Mamba marking its first application to PCa detection in bpMRI. While MTL has been explored before, our work is the first to use zonal anatomy as auxiliary segmentation targets to improve lesion detection in bpMRI.  We do agree that the paper type should be application, and we agree to change if possible.

---

### **R1: Failure cases are not demonstrated.**
Figure 2 now includes failure cases, highlighting false positives due to ADC hypointensity and false negatives from ground truth inconsistencies.

---

### **R1: The computational burden is unclear.**
Parameter counts, training time, and inference time are now provided in Table 2 (main text) and Table 8 (appendix).

---

### **R1: Is there external validation beyond PI-CAI?**
Yes, we validate on an out-of-distribution in-house dataset (N=200, St. Olavs Hospital).

---

### **R2: Why train for only 200 epochs?**
Early experiments showed convergence around 100 epochs, with overfitting beyond 200 epochs. This is now clarified in the methodology section.

---

### **R3: Who performed the in-house dataset annotations?**
Annotations followed PI-CAI standards. A resident radiologist (≥2 years experience) and a senior radiologist (≥10 years experience) performed the delineations.

---

### **R3: Why does nnDetection predict square-like lesions?**
nnDetection only produces bounding boxes due to its detection nature. This is now clarified in the discussion.

---

### **R3: Which model weight was used for testing?**
A mean ensemble across all model fold predictions was used. This is clarified in the methodology.

**Supporting Material:**

/attachment/7f3a957c57c8f9c4a9922034c5a77210d109a419.zip

---

### Meta-Review · Area_Chair_QvWa · 2025-03-21

**Recommendation:** Reject
**Confidence:** 4

**Metareview:**

The primary concern across all three reviewers' centers on the limited methodological novelty of this work. While the application of U-Mamba to prostate cancer detection and the integration of multi-task learning (MTL) are explored, reviewers consistently point out that both U-Mamba and MTL techniques are increasingly common in medical imaging. I agree with reviewers that the paper leans more towards an application paper than a technical innovation and the core contribution is essentially an evaluation of U-Mamba on a new task, with MTL being a well-established technique in this domain. The combination of U-Mamba with auxiliary zonal segmentation, while producing good results, lacks significant novelty. Though the authors have attempted to address these concerns in their rebuttal by highlighting the specific combination of zonal anatomy-guided MTL with U-Mamba in bpMRI, the fundamental issue of incremental advancement remains. Therefore, due to the limited novelty, I recommend rejection.